# Deciphering Complex Interactions in Bioactive Lipid Signaling

**DOI:** 10.3390/molecules28062622

**Published:** 2023-03-14

**Authors:** Mauro Maccarrone

**Affiliations:** 1Department of Biotechnological and Applied Clinical Sciences, University of L’Aquila, 67100 L’Aquila, Italy; mauro.maccarrone@univaq.it; 2European Center for Brain Research (CERC), Santa Lucia Foundation IRCCS, 00143 Rome, Italy

**Keywords:** biased signaling, eicosanoid, endocannabinoid, metabolism, signal transduction, specialized pro-resolving mediator, sphingosine-1-phosphate, steroid

## Abstract

Lipids are usually viewed as metabolic fuel and structural membrane components. Yet, in recent years, different families of lipids able to act as authentic messengers between cells and/or intracellularly have been discovered. Such lipid signals have been shown to exert their biological activity via specific receptors that, by triggering distinct signal transduction pathways, regulate manifold pathophysiological processes in our body. Here, endogenous bioactive lipids produced from arachidonic acid (AA) and other poly-unsaturated fatty acids will be presented, in order to put into better perspective the relevance of their mutual interactions for health and disease conditions. To this end, metabolism and signal transduction pathways of classical eicosanoids, endocannabinoids and specialized pro-resolving mediators will be described, and the intersections and commonalities of their metabolic enzymes and binding receptors will be discussed. Moreover, the interactions of AA-derived signals with other bioactive lipids such as shingosine-1-phosphate and steroid hormones will be addressed.

## 1. Introduction

Lipids are usually viewed as metabolic fuel and structural membrane components. Yet, in recent years, different families of lipids able to act as authentic messengers between cells and/or intracellularly have been discovered and have been shown to exert their biological activity by binding to specific receptors, thus triggering signal transduction pathways that regulate manifold pathophysiological processes in our body.

In general, bioactive lipids in cell signaling include plant-derived compounds such as cannabinoids (terpeno-phenol compounds also known as phytocannabinoids) [1,2], terpenoids (including sterols) [3,4], carotenoids and phenolics (including flavonoids) [5,6]. In addition, our cells produce lipids endowed with biological activity in signal transduction, the best characterized being derivatives of poly-unsaturated fatty acids (PUFAs). These endogenous substances support a complex network of molecular and cellular events responsible for a plethora of physiopathological conditions such as inflammation and chronic diseases, cancer, diabetes, atherosclerosis, asthma, autoimmune and neurodegenerative disorders [7], as well as reproduction [8,9], just to mention a few.

To date, the three major classes of endogenous bioactive lipids that have been best characterized are classical eicosanoids (EICs), endocannabinoids (ECBs) and specialized pro-resolving mediators (SPMs) [7]. A PubMed search conducted over the last 30 years clearly shows that interest in these substances has always remained high for EICs, the first to be discovered in the 1930s [10], and has been continuously growing for ECBs and SPMs since their first identification in 1992 [11] and 1974 [12], respectively (Table 1).

In order to put into better perspective the relevance of the potential interactions of different bioactive lipids for health and disease conditions, the metabolism and signal transduction pathways of these compounds are summarized below.

## 2. Bioactive Lipids

### 2.1. Eicosanoids

Eicosanoids (EICs) are a family of ~120 biologically active compounds derived from unsaturated fatty acids, among which n-6 (ω-6) arachidonic acid (eicosatetraenoic acid, C20:4) is the main precursor. The history of the EICs dates back to the 1930s, when new biologically active lipids were found in human seminal plasma [10]. Since then, they have been shown to play a major role in key processes like inflammation [13] and maintenance of energy homeostasis [14,15,16,17].

Upon specific stimuli, phospholipase A_2_ (PLA_2_) releases arachidonic acid (AA) from the *sn*-2 position of membrane glycerophosholipids [18,19], and then EICs are generated through three main biosynthetic pathways, driven by (i) cyclooxygenases (COXs) [20], (ii) lipoxygenases (LOXs) [21], and (iii) cytochrome P450 epoxydases (CYP450s) [22].

Cyclooxygenase-1 (COX-1) and -2 (COX-2) generate EICs that have a ring in their structure and are collectively termed prostanoids; the latter include prostaglandins (PGs), prostacyclins (PGIs) and thromboxanes (TXs) that are all formed from the common precursor prostaglandin H_2_ (PGH_2_). All prostanoids bind to and activate distinct G protein-coupled receptors (GPCRs), as summarized in Figure 1, and thus, they exert well-recognized hormone-like actions on the surrounding cells or the producing cells in a paracrine or autocrine manner [20]. Consequently, prostanoids trigger signal transduction on various human cells such as neuron, retina, vessel, heart, and blood cells.

Different lipoxygenase (LOX) isozymes such as 5-LOX, 12-LOX and 15-LOX promote the conversion of AA into hydroperoxy derivatives (hydroperoxyeicosatetraenoic acids, HpETEs). 5-HpETE is a precursor of leukotrienes LTA_4_ and LTB_4_, as well as of cysteinyl-LTs (CysLTs) such as LTC_4_, LTD_4_ and LTE_4_; 12-HpETE is a precursor of hepoxilins (HxA and HxB); and 15-HpETE is a precursor of lipoxins (LXA_4_ and LXB_4_) [21]. Of note, due to their pro-resolving properties, lipoxins are generally also listed among SPMs, detailed in the following Section 2.3. Much like COX-generated compounds, LOX-derived EICs bind to distinct GPCRs, some of which have been identified. LTB_4_ binds to and activates BLT1 and BLT2 receptors, whereas CysLTs bind to and activate CyLTR1 and CysLTR2 receptors [23].

Finally, CYP2C and CYP2J are two distinct CYP450s that produce epoxyeicosatetraenoic acids (EETs), whereas CYP4A and CYP4F convert AA into 20-hydroxides (20-HETE) [22], as schematically depicted in Figure 1.

### 2.2. Endocannabinoids

The endocannabinoids (ECBs) bind to and activate the same cannabinoid 1 (CB_1_) and 2 (CB_2_) GPCRs activated by Δ^9^-tetrahydrocannabinol (THC), the main psychoactive component of cannabis (*Cannabis sativa* and *Cannabis indica*) [24,25]. Therefore, ECBs exert a wide range of pathophysiological functions, both in the central nervous system and in peripheral tissues, generally acting as pro-homeostatic modulators [26]. The main ECBs so far identified are *N*-arachidonoylethanolamine or anandamide (AEA), which is an amide of AA with ethanolamine [11], and 2-arachidonoylglycerol (2-AG), which is an ester of AA with glycerol [27]. AEA is produced mainly via *N*-acylphosphatidylethanolamines-specific phospholipase D (NAPE-PLD) [28], and 2-AG mainly via *sn*-1-diacylglycerol lipases (DAGLs) α and β [29]. Yet, additional biosynthetic enzymes can contribute to AEA and 2-AG production, such as cytosolic phospholipase A_2γ_ [30], phospholipase C [31] and the αβ-hydrolase domain 4 (ABHD4) protein [32]. Then, degradation of AEA is primarily catalyzed by fatty acid amide hydrolase (FAAH), which cleaves it into AA and ethanolamine [33]. Instead, 2-AG is cleaved by monoacylglycerol lipase (MAGL) that releases AA and glycerol [34]. Much like the biosynthesis, the degradation of AEA and 2-AG also can occur via additional hydrolases or can be paralleled by alternative pathways such as oxygenation through COX-2, 12-LOX, 15-LOX, or CYP450s [35]. The latter oxygenases generate ECB-like derivatives known as PG-ethanolamides, glyceryl esters, hydroxy-AEAs and hydroxy-eicosatetraenoylglycerols. These oxidized compounds seem to have their own biological activity by activating distinct receptor targets, yet to be identified [26,36]. Indeed, both AEA and 2-AG bind to and activate G_i/o_ protein-coupled CB_1_ and CB_2_ receptors [37,38], along with additional ECB-binding receptors, such as the orphan G protein-coupled receptors GPR55 and GPR119, the transient receptor potential vanilloid type-1 (TRPV1) ion channel, and the nuclear peroxisome proliferator-activated receptors (PPAR) α, γ and δ [25]. Thus, AEA and 2-AG, in addition to congeners such as *N*-archidonoyldopamine (NADA) and *N*-oleoyldopamine (OLDA), are also considered bona fide endovanilloids [39]. Incidentally, there are ECB-like compounds, such as *N*-palmitoylethanolamine (PEA) and *N*-oleoylethanolamine (OEA), and amphiphilic molecules, such as *N*-acyl amino acids (e.g., *N*-arachidonoylglycine), that also share enzymatic pathways and/or interactions with receptors of the ECBs, thus modulating various physiological and pathological conditions [25]. 

The metabolic pathways and known receptors of eCBs, that together all form the so-called “endocannabinoid system” are summarized in Figure 2.

### 2.3. Specialized Pro-Resolving Mediators

The specialized pro-resolving mediators (SPMs) had been identified in 1974 as anti-inflammatory lipids [12]. Indeed, their name refers to their major role as drivers of resolution of inflammation. Unlike classical EICs that are produced mainly from n-6 (ω-6) AA, almost all SPMs derive from n-3 (ω-3) PUFAs, with only a few compounds sharing with EICs the n-6 (ω-6) AA precursor [40]. Depending on the PUFAs used for the synthesis, SPMs are classified as derivatives of (i) docosahexaenoic acid (DHA), (ii) docosapentaenoic acid (DPA), (iii) eicosapentaenoic acid (EPA), and (iv) AA. 

The biosynthesis of SPMs resembles that of EICs, because it starts with the PLA_2_-catalyzed release of PUFAs from membrane phospholipids, followed by further processing via COX-2 and/or various LOX and CYP450 isozymes. 

DHA-derived SPMs are the largest class of SPMs and include heterogeneous D-series resolvins 1–6 (RvD 1–6), maresins 1,2 (MaR1 and MaR2), and neuroprotectins/protectins 1, X (PD1/NPD1 and PDX). 12-LOX or 15-LOX generate the 14(*S*)- or 17(*S*)-hydroperoxides of DHA (14(*S*)-HpDHA and 17(*S*)-HpDHA), respectively. Then, maresins are obtained through further metabolism of 14(*S*)-HpDHA, whereas RvDs and PDs/NPDs are generated from 17(*S*)-HpDHA. To date, the following GPCRs have been identified as targets of DHA-derived SPMs: formyl peptide receptor 2 (FPR2)—also known as ALX (FPR2/ALX)—a target of RvD1 and RvD3; GPR18, a target of RvD2; GPR32, a target of RvD1, RvD3 and RvD5; GPR37, a target of PD1; and Leucine-rich repeat-containing G protein-coupled receptor 6 (LGR6), a target of MaR1 [40]. 

DPA-derived SPMs are a newly discovered class of SPMs that include 13-series resolvins (RvTs) as well as the DPA-derived counterparts of DHA SPMs. The metabolic enzymes that convert DPA into SPMs are, again, COX-2, 12-LOX and 15-LOX [41,42].

EPA-derived SPMs are a small group that includes E-series resolvins 1–3 (RvE1–3). Upon release of EPA from membrane phospholipids, COX-2 or CYP450 promote its conversion to 18(*R/S*)-hydroperoxyeicosapentaenoic acid (18(*R/S*)-HpEPE) epimers that are then further metabolized through 5/12/15-LOX, peroxidases and hydrolases. As yet, only one receptor has been shown to be activated by RvEs, and it is termed the chemerin receptor 23 (ChemR23). Yet, LTB_4_-binding BLT1 receptor also can be antagonized by both RvE1 and RvE2 [43].

Finally, AA-derived SPMs are the same lipoxins (LXA_4_ and LXB_4_) described above as members of the classical EICs. Both LXA_4_ and LXB_4_ bind to and activate FPR2/ALX, a GPCR [44]. The metabolic pathways and known receptors of SPMs are summarized in Figure 3.

## 3. Interactions in Lipid Signaling Pathways

### 3.1. Common Metabolic Pathways

From the metabolic pathways of EICs, ECBs and SPMs shown in Figure 1, Figure 2 and Figure 3, it is apparent that these families of bioactive lipids share key enzymes, such as the lipase responsible for the release from phospholipid precursors (PLA_2_) and three oxygenases able to generate from AA: (i) epoxyeicosatrienoic acids (CYP450); (ii) 12-hydroxyeicosatetraenoic acids (12-LOX); and (iii) cyclic prostanoid(-like) oxidative derivatives (COX-2). Of note, PLA_2_ belongs to a superfamily of enzymes that play major roles in several pathophysiological processes, e.g., mast cell biology [45], atherosclerotic cardiovascular disease [46], and gastrointestinal diseases [47]. Unsurprisingly, PLA_2_ enzymes are able to mobilize unique bioactive lipids in multiple ways that are spatiotemporally regulated, thus triggering distinct signaling pathways. In addition, an emerging concept in the control of the synthesis of oxygenated metabolites of AA—as well as other PUFAs—is the compartmentalization of the enzymes and lipid substrates within the cell, a clue that allows shaping the appropriate biological response in different cellular contexts [48]. Unfortunately, the molecular mechanisms that underlie the fine tuning of these enzymes, and their ability to drive signaling through one class of lipids or another, remain elusive. Yet, substrate hydrolysis has been carefully interrogated in the case of COX-2, by comparing oxygenation of AA to prostaglandins (PGs) and of the AA ester 2-AG to prostaglandin glyceryl esters (PG-Gs) [49]. The efficiency of in vitro oxygenation of these two substrates by COX-2 has been found to be similar, but competition between AA and 2-AG at the catalytic and allosteric sites of the enzyme (a dimer), combined with a differential allosteric modulation, leads to preferential oxygenation of AA in vivo [49]. This is just one example, thoroughly investigated, of the complex interplay between different bioactive lipids and their metabolic enzymes, with major functional consequences for the engagement of downstream targets that in turn trigger a biological activity [35]. Common enzymes between EICs, ECBs and SPMs are summarized in Table 2.

Besides enzymes, protein carriers [50] or microvesicles [51] that transport bioactive lipids inside and outside the cell can also be expected to contribute to fueling one metabolic route rather than another, as it is becoming apparent in the case of ECBs [52,53,54]. Unfortunately, for known lipid transporters such as heat shock protein 70 and albumin [55], fatty acid-binding proteins 1, 5 and 7 [56,57], FAAH-like anandamide transporter [58], sterol carrier protein 2 [59] and retinol-binding protein 2 [60], the mechanisms that support the choice of one substrate over another also remain to be elucidated. At any rate, crosstalk among different bioactive lipids are emerging in the literature. For instance, RvD1 has been shown to produce potent antiallodynia and antihyperalgesia in a model of bone cancer pain by also triggering spinal upregulation of ECBs, which produce additional antinociception predominantly through CB_2_ receptors [61]. Moreover, activation of the DHA-derived SPM-binding receptor GPR18 has been shown to drive neuroprotection against HIV-1 Tat-induced neurodegeneration by engaging FAAH, the key enzyme for AEA hydrolysis [62]. The same GPR18 receptor has been also shown to modulate ECB signaling in metabolic dysfunction, as well as in microglia [63]. Incidentally, it should be noted that more subtle interactions may also occur directly between lipid metabolic enzymes, as in the case of 5-LOX that blocks FAAH activity in human mast cells by producing AEA hydroperoxides that act as enzyme inhibitors [64].

### 3.2. Common Receptor Targets

It is remarkable that almost all receptors of bioactive lipids are G protein-coupled receptors (GPCRs), the exceptions being the ECB-binding TRPV1 channels and PPARα/γ/δ nuclear receptors (Figure 1, Figure 2 and Figure 3). GPCRs are one of the most important classes of membrane receptors, able to sense an extraordinary variety of biomolecules in order to activate multiple intracellular signaling cascades [65]. An unresolved issue about GPCRs is signal bias (also known as biased signaling, biased agonism or functional selectivity), whereby a ligand-dependent selectivity exists for certain signal transduction pathways compared to a reference ligand (often the endogenous agonist) at the same receptor. Biased agonists for GPCRs are of profound pharmacological interest, because they could usher in a new generation of drugs with greatly reduced side effects [66]. An interesting example of the alternative pathways that can be activated by different ligands at the same receptor, and by the same ligand at different receptors, is represented by ECB signaling [25], schematically depicted in Figure 4.

Unlike metabolic pathways that share some of the enzymes responsible for synthesis and degradation (Table 2), receptors seem to be clearly distinct for EICs, ECBs and SPMs (Figure 1, Figure 2 and Figure 3). However, emerging evidence supports the possibility that one class of bioactive lipids can bind to and activate specific receptors of another class, adding a further dimension to the complexity of the transduction pathways of these molecules. Indeed, the endogenous ECB-like compound *N*-arachidonoylglycine has been recently shown to activate the SPM-binding receptor GPR18 with neuroprotective effects [67], and so does the plant-derived substance cannabidiol, with an impact on Alzheimer’s disease [68] and, apparently more in general, on the balance between different lipid signals. Additionally, the endogenous ECB-like compound, *N*-palmitoylethanolamine (PEA), has been shown to promote a pro-resolving macrophage phenotype [69], and selective activation of CB_2_ improves efferocytosis, which is a typical hallmark of resolution of inflammation [70]. Therefore, GPR18 and CB_2_ seem to be a cross-roads for the action of ECBs and SPMs, apparently also due to the possible formation of heterodimers [71]. GPR18-CB_2_ heteroreceptor complexes display particular functional properties that often consist of negative crosstalk (i.e., activation of one receptor reduces signaling that arises from the partner receptor) and cross-antagonisms (i.e., the response of one of the receptors is blocked by a selective antagonist of the partner receptor). Once again, this is just one example of the complexity of the crosstalk between different classes of bioactive lipids, which might contribute to biased signaling triggered by each of them [72]. 

Additionally, lipoxin A_4_ (LXA_4_), an anti-inflammatory EIC/SPM, has been shown to act as an endogenous allosteric enhancer of CB_1_ receptor [73], able to reinforce the biological activity of AEA [74]. Moreover, the peroxidation product 12(*S*)-hydroxyeicosatetraenoic acid, generated by 12-LOX from AA (Figure 1), has been shown to mediate diabetes-induced endothelial dysfunction by activating TRPV1 [75], leaving open the possibility that the latter receptor channel may be a target not only of ECBs, but also of EICs and SPMs. In this context, LTB_4_-binding BLT1 receptor can be antagonized by both RvE1 and RvE2 [43], thus bridging together the actions of EICs and SPMs (Table 2).

On a final note, it should be stressed that a pre-requisite to design effective and selective drugs able to modulate endogenous lipid signaling is the clear understanding of the 3D structures of the molecular targets. This knowledge is largely available for the metabolic enzymes and binding receptors of EICs, a lipid class known for quite a few decades [10]. Surprisingly, for the molecular machinery that drives ECB signaling, to date, 3D structures of only 23 major components have been resolved, whereas many other receptors (e.g., GPR55, GPR119 and TRPV4), enzymes (e.g., NAT, DAGLα/β, GDE1,4,7, ABHD2,4,6,12) and putative membrane transporters still await clarification [26]. Finally, for SPMs, only the 3D structures of formyl peptide receptor 2 (FPR2) [76] and leukotriene B_4_ receptor 1 (BLT1) [77] have been recently elucidated, making it difficult to thoroughly investigate the mechanisms of binding, activation and regulation of these GPCRs. It is apparent that such an information gap is particularly troubling for drug discovery programs and must be urgently filled.

### 3.3. Interaction of EICs, ECBs and SPMs with AA/PUFA-Unrelated Bioactive Lipids

In addition to the above-discussed interactions among AA/PUFA-derived EICs, ECBs and SPMs, additional bioactive lipids can crosstalk with them. For instance, sphingolipids are a relevant class of bioactive lipids that play key roles in the regulation of several cellular processes, including growth, differentiation, inflammatory responses, and apoptosis. Recent studies implicated a regulatory function of sphingolipids in prostaglandin production, and sphingosine-1-phosphate (S1P) has been shown to induce COX-2 expression [78]. In line with this, S1P has been recently demonstrated to activate TRPV1, with an impact on AEA-dependent regulation of mitochondrial activity [79]; conversely, RvD1 activates the S1P signaling pathway [80]. Moreover, steroid hormones also can contribute to modulating AA/PUFA-derived lipid signaling, for instance, at the level of gene expression. Such a regulation has been indeed shown for estrogen that enhances FAAH expression via an epigenetic mechanism mediated by histone demethylase LSD1 [81], and for progesterone that activates the promoter of the same AEA hydrolase through the transcription factor Ikaros [82]. These findings open new perspectives for the development of epigenetic drugs (epidrugs) able to modulate bioactive lipid signaling [83]. Overall, these data support a further layer of complexity in bioactive lipid crosstalk, well beyond the family of AA/PUFA-derived molecules.

## 4. Conclusions and Future Directions

In recent years, a diverse set of signaling lipids has been demonstrated to interact with specific receptors under a tight metabolic control, with an impact on many pathophysiological processes. Unlike protein signals that can be stored in vesicles, lipid messengers are often synthesized “on demand”, i.e., when and where needed, and their endogenous content is tightly regulated by distinct biosynthetic and hydrolytic enzymes. Understanding mutual regulations of these enzymes, possibly through post-translational modifications, protein–protein and protein–lipid interactions, as well as via membrane and subcellular location, now seems urgent in order to exploit lipid signals as potential next-generation therapeutics. In particular, it should be recalled that location is a critical element to modulating metabolic enzymes and receptors, and hence to driving signaling. In line with this, the possibility of visualizing intracellular trafficking of lipid-interacting proteins seems relevant, making necessary and urgent the development and use of chemical probes. Only recently have the development of small-molecule inhibitors and activity-based probes, along with the use of advanced techniques like activity-based protein profiling and chemical proteomics, been essential to guide the drug discovery and development of compounds able to target lipid signaling, for instance allowing to visualize key elements of ECB signaling, such as FAAH and MAGL hydrolases [84] and CB_2_ receptor [85]. Now that tools for more accurately tracking lipid metabolism, location, and action have been provided, and that innovations in bioimaging techniques have been made, it can be anticipated that the underlying mechanisms of lipid signaling and of the crosstalk among different lipid classes will be uncovered at nanoscale resolution, thus allowing to interrogate lipid networks in real life.

## Figures and Tables

**Figure 1 molecules-28-02622-f001:**
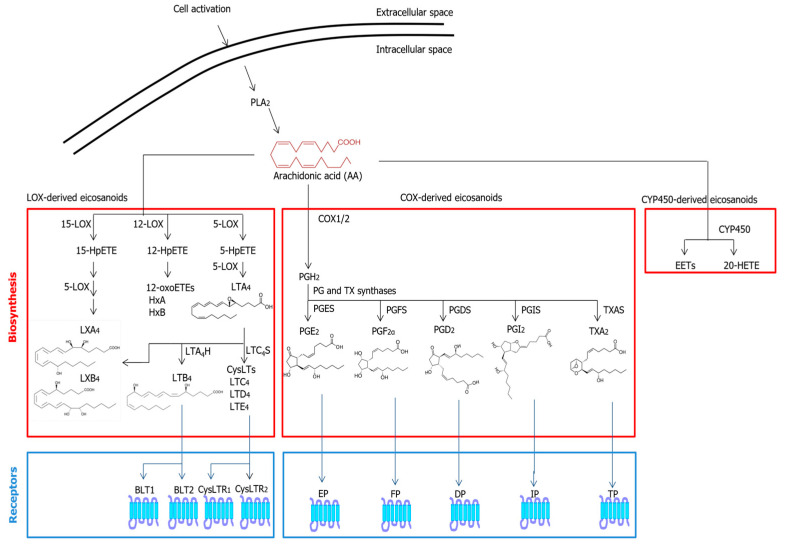
Metabolic pathways and known receptors of classical EICs. Abbreviations: 20-HETE—20-hydroxyeicosatetraenoic acid; BLT—leukotriene B_4_ receptor; COX—cyclooxygenase; CYP450—cytochrome P450; DP—prostaglandin D receptor; EET—epoxyeicosatrienoic acid; EP—prostaglandin E receptor; FP—prostaglandin F receptor; HpETE—hydroperoxyeicosatetraenoic acid; IP—prostacyclin I receptor; LOX—lipoxygenase; LT—leukotriene; LX—lipoxin; PG—prostaglandin; PGDS—prostaglandin D_2_ synthase; PGES—prostaglandin E synthase; PGFS—prostaglandin F synthase; PGI_2_—prostacyclin 2; PGIS—prostacyclin I synthase; PLA_2_—phospholipase A_2_; TP—thromboxane receptor; TXAS—thromboxane synthase.

**Figure 2 molecules-28-02622-f002:**
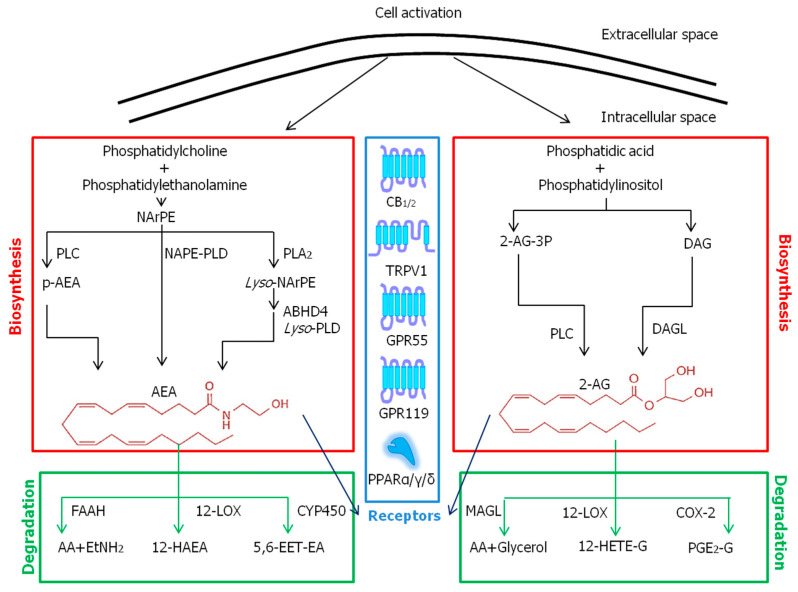
Metabolic pathways and known receptors of ECBs. Abbreviations: 12-HAEA—12-hydroxy-*N*-arachidoylethanolamine; 12-HETE-G—12-hydroxy eicosatetraenoylglycerol; 2-AG—2-arachidonoylglycerol; 5,6-EET-EA—5,6-epoxyeicosatrienoyl ethanolamide; AA—arachidonic acid; ABHD4—αβ-hydrolase domain 4 protein; AEA—*N*-arachidonoylethanolamine; CB_1/2_—cannabinoid receptor 1/2; COX-2—cyclooxygenase-2; CYP450—cytochrome P450; DAG—diacylglycerol; DAGL—diacylglycerol lipase; EtNH_2_—ethanolamine; FAAH—fatty acid amide hydrolase; GPR—G protein-coupled receptor; LOX—lipoxygenase; Lyso-NArPE—*lyso*-*N*-arachidonoyl phosphatidylethanolamine; Lyso-PLD—*lyso*phospholipase D; MAGL—monoacylglycerol lipase; NAPE-PLD—*N*-acylphosphatidylethanolamines-specific phospholipase D; NArPE—*N*-arachidonoyl phosphatidylethanolamine; p-AEA—phospho-*N*-arachidonoylethanolamine; PGE_2_-G—prostaglandin E_2_ glycerol; PLA_2_—phospholipase A_2_; PLC—phospholipase C; PPARα/γ/δ—peroxisome proliferator activated receptor α/γ/δ; TRPV1—transient receptor potential vanilloid type-1.

**Figure 3 molecules-28-02622-f003:**
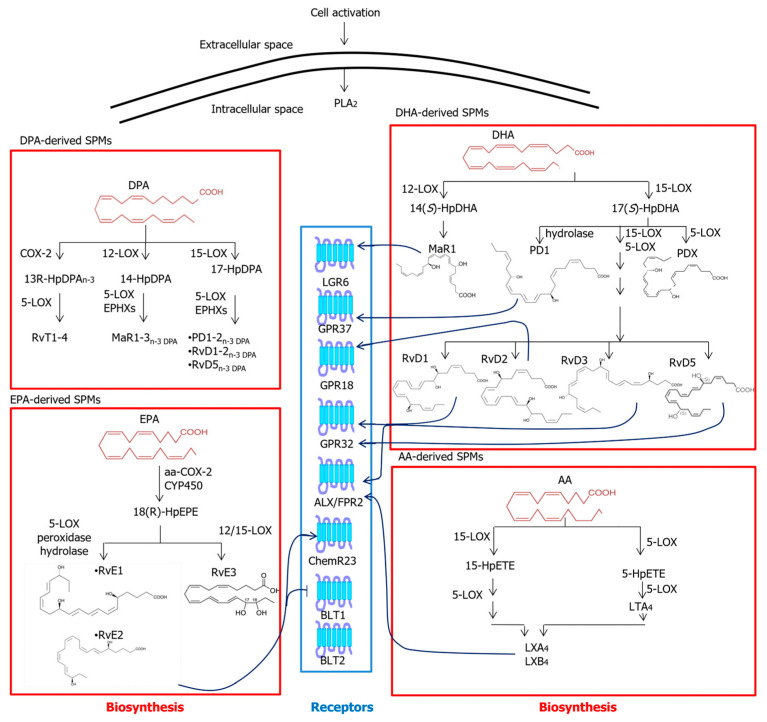
Metabolic pathways and known receptors of SPMs. Abbreviations: AA—arachidonic acid; ALX/FPR2—formyl peptide receptor 2; BLT—leukotriene B_4_ receptor; ChemR23—chemerin receptor 23; COX—cyclooxygenase; CYP450—cytochrome P450; DHA—docosahexaenoic acid; EPA—eicosapentaenoic acid; EPHX—epoxide hydrolase; GPR—G protein-coupled receptor; HpDHA—hydroperoxy-docosahexaenoic acid; HpDPA—hydroperoxydocosapentaenoic acid; HpEPE—hydroperoxyeicosapentaenoic acid; HpETE—hydroperoxyeicosatetraenoic acid; LGR6—leucine-rich repeat-containing G protein-coupled receptor 6; LOX—lipoxygenase; LT—leukotriene; LX—lipoxin; MaR1—maresin 1; n-3 DPA—n-3 docosapentaenoic acid; PD1—protectin D1; PDX—protectin DX; PLA_2_—phospholipase A_2_; RvD—D-series resolvins; RvE—E-series resolvins; RvTs—13-series resolvins.

**Figure 4 molecules-28-02622-f004:**
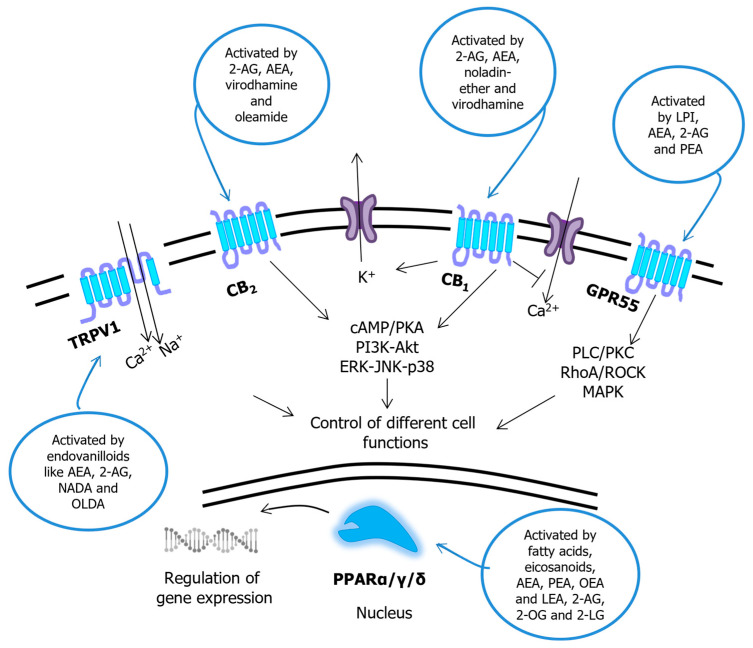
Signal transduction pathways triggered by endocannabinoids and congeners. Abbreviations: AEA—*N*-arachidonoylethanolamine (anandamide); 2-AG—2-arachidonoylglycerol; 2-LG—2-linoleoylglycerol; PEA—*N*-palmitoylethanolamine; LEA—*N*-linoleoylethanolamine; OEA—*N*-oleoylethanolamine; 2-OG—2-oleoyglycerol; LPI—lysophosphatidic acid; NADA—*N*-arachidonoyldopamine; OLDA—*N*-oleoyldopamine; PLC—phospholipase C; PKC—protein kinase C; ROCK—Rho-associated protein kinase; MAPK—mitogen-activated protein kinase; cAMP—cyclic AMP; PKA—protein kinase A; PI3K—phosphatidylinositol (phosphoinositide) 3-kinase; Akt—α-serine/threonine-protein kinase; ERK—extracellular signal-regulated kinase; JNK—c-Jun N-terminal kinase; GPR55—G protein-coupled receptor 55; CB_1/2_—cannabinoid receptor 1/2; TRPV1—transient receptor potential vanilloid 1. Modified from ref. [25].

**Table 1 molecules-28-02622-t001:** Number of entries of a PubMed search with “eicosanoids”, “endocannabinoids” and “specialized pro-resolving mediators” over the last 30 years.

Time Range	EICs	ECBs	SPMs
1992–1997	22,258	200	3
1998–2002	16,534	648	5
2003–2007	17,065	1791	18
2008–2012	16,128	2866	85
2013–2017	14,402	3271	316
2018–2022	10,713	3949	847

Abbreviations: EICs, eicosanoids; ECBs, endocannabinoids; SPMs, specialized pro-resolving mediators.

**Table 2 molecules-28-02622-t002:** Common metabolic enzymes and receptors of different classes of bioactive lipids.

Enzyme/Receptor	EICs	ECBs	SPMs
PLA_2_	+	+	+
CYP450	+	+	+
12-LOX	+	+	+
COX-2	+	+	+
CB_1_	+	+	+
CB_2_		+	+
GPR18		+	+
TRPV1	+	+	+
BLT1	+		+

Abbreviations: EICs—eicosanoids; ECBs—endocannabinoids; SPMs—specialized pro-resolving mediators; PLA_2_—phospholipase A_2_; CYP450—cytochrome P450; 12-LOX—12-lipoxygenase; COX-2: cyclooxygenase-2; CB_1/2_—cannabinoid receptor 1/2; GPR18—G protein-coupled receptor 18; TRPV1—transient receptor potential vanilloid-1; BLT1—leukotriene B_4_ receptor.

## Data Availability

Data sharing not applicable.

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
