# Peer review of "Deciphering Complex Interactions in Bioactive Lipid Signaling"

_molecules, 2023, doi:10.3390/molecules28062622_

Round 1

Reviewer 1 Report

The expanding field of bioactive lipids has generated growing interest in their role within various physiopathological conditions and metabolic signaling pathways. The Author elegantly describes and elaborates on the three main endogenous bioactive lipids; the classical eicosanoids (EICs), endocannabinoids (ECBs), and specialized pro-resolving mediators (SPMs), their biosynthesis and degradation enzymatic pathways and receptors as well as the interactions between their signaling pathways and shared enzymes. The author specifically underscores the complexity of the signaling pathways of these three endogenous bioactive lipids, in terms of common enzymes, receptors, and ligands as well as the interaction of other non-arachidonic acid (AA)/PUFAs-derived bioactive lipids. Moreover, he emphasizes the need in understanding the three-dimensional structure and visualization of various components of endogenous lipid signaling as critical for the development of more precise and selective drugs. The manuscript is well designed, written and concluded. I recommend acceptance of this article after addressing the following minor comments:

1.     In section 3.3, I would suggest adding the amphiphilic molecules, N-acyl amino acids (NAAs), which also share enzymatic pathways and interaction with receptors of the bioactive lipids described. See PMID: 34217720 and 31817019. This is particularly important since N-arachidonoylglycine (for example) is mentioned in the text (see page 9 line 267), but its relation to any of the well-documented groups is not described. Also, like N-arachidonoylglycine, many other N-arachidonoyl-amino acids have been shown to modulate various physiological and pathological conditions.

2.     Page 2, line 62. Please define AA, as it is described in the text for the first time.

3.     Page 2, line 75. LXA4: As defined later in the text, this molecule has been shown to activate the formyl peptide receptor 2 (FPR2) - also known as ALX (FPR2/ALX) (PMID: 27072607). See also PMID:17767357. It has also been shown that LXA4 modulate the CB1R (PMID: 23150578). This information should be added here and presented in Figure 1 too.

4.     Page 9, lines 269-270. Since the focus of this manuscript is to highlight endogenous bioactive compounds, I suggest omitting the sentence describing the role of cannabidiol in AD.

5.     The author should consider briefly describing the role of PEA and OEA, as they are highlighted in Figure 4.

6.     Page 9, line 282. Lipoxin A4 needs to be defined here as LXA4.  

Author Response

Replies to Referee n. 1

First of all, I like to thank the Referee for her/his kind appreciation of my work, and for constructive comments that helped indeed to improve its clarity and impact. Point by point replies to the criticisms raised are detailed below.

  1. In section 3.3, I would suggest adding the amphiphilic molecules, N-acyl amino acids (NAAs), which also share enzymatic pathways and interaction with receptors of the bioactive lipids described. See PMID: 34217720 and 31817019. This is particularly important since N-arachidonoylglycine (for example) is mentioned in the text (see page 9 line 267), but its relation to any of the well-documented groups is not described. Also, like N-arachidonoylglycine, many other N-arachidonoylamino acids have been shown to modulate various physiological and pathological conditions.

R:   NAAs have been included in the amended text, as suggested. See page 4, lines 132-135. 

  1. Page 2, line 62. Please define AA, as it is described in the text for the first time.

R:   Defined as requested. See page 2, line 64. 

  1. Page 2, line 75. LXA4: As defined later in the text, this molecule has been shown to activate the formyl peptide receptor 2 (FPR2) - also known as ALX (FPR2/ALX) (PMID: 27072607). See also PMID:17767357. It has also been shown that LXA4 modulate the CB1R (PMID: 23150578). This information should be added here and presented in Figure 1 too.

R:   The allosteric modulation of CB1 by LXA4 has been discussed, and the suggested reference has been quoted as ref. 73. See page 9, lines 312-314.

  1. Page 9, lines 269-270. Since the focus of this manuscript is to highlight endogenous bioactive compounds, I suggest omitting the sentence describing the role of cannabidiol in AD.

R:   I have to disagree with this suggestion, because cannabidiol seems to have a particular relevance in driving bioactive lipid signaling. This has been better clarified in the revised text (page 9, lines 299-300). I hope the Referee can accept this point.

  1. The author should consider briefly describing the role of PEA and OEA, as they are highlighted in Figure 4.

R:   Done as suggested. See page 4, lines 131-132.

  1. Page 9, line 282. Lipoxin A4 needs to be defined here as LXA4.

R:   Done as requested. See page 9, line 312.

Reviewer 2 Report

The manuscript molecules-2271776 “Deciphering Complex Interactions in Bioactive Lipid Signaling” by M. Maccarrone presents an overview on lipids involved in intracellular signaling.  The manuscript provides useful and interesting information and, in my opinion, can be published after some adjustments.

1. One main observation regarding the manuscript is the very limited discussion regarding the role of prostanoids (in eicosanoids class) as hormone like compounds.  A more detailed discussion on their role must be included.

2. Another observation is that across the manuscript the word “bioactive” is used with the restricted sense of “bioactive in intercellular signaling”.  The role of lipids as metabolic fuel is still a ”bioactive” function.  A clear distinction must be made between the naming of different classes of lipids and to specify “bioactive in intercellular signaling” in places such as Section 2.

3. Table 2 should have the title : “Number of entries…” and not “Results...”  Also, in English speaking countries the decimals are indicated with a dot “.” and the thousands are separated with a coma “,”.  This should be corrected in Table 2.

Author Response

Replies to Referee n. 2

First of all, I like to thank the Referee for her/his kind appreciation of my work, and for constructive comments that helped indeed to improve its clarity and impact. Point by point replies to the criticisms raised are detailed below.

  1. One main observation regarding the manuscript is the very limited discussion regarding the role of prostanoids (in eicosanoids class) as hormone-like compounds. A more detailed discussion on their role must be included.

R: A more detailed description of the hormone-like activity of prostanoids has been included, as requested. See page 2, lines 73-76.

  1. Another observation is that across the manuscript the word “bioactive” is used with the restricted sense of “bioactive in intercellular signaling”. The role of lipids as metabolic fuel is still a ”bioactive” function. A clear distinction must be made between the naming of different classes of lipids and to specify “bioactive in intercellular signaling” in places such as Section 2.

R: The requested distinction has been made more clearly, as suggested. See page 1, lines 31, 34-35.

  1. Table 2 should have the title : “Number of entries…” and not “Results...” Also, in English speaking countries the decimals are indicated with a dot “.” and the thousands are separated with a coma “,”. This should be corrected in Table 2.

R: Table 1 has been amended as suggested. See page 2.
